# Perceived Telehealth Usability for Personalized Healthcare Among the Adult Population in Saudi Arabia: A Cross-Sectional Study in the Post-COVID-19 Era

**DOI:** 10.3390/healthcare13010062

**Published:** 2025-01-01

**Authors:** Samiha Hamdi Sayed, Danyah Abdulrahman Aldailami, Maha Mahmoud Abd El Aziz, Ebtesam Abbas Elsayed

**Affiliations:** 1Public Health Department, College of Health Sciences, Saudi Electronic University, Riyadh 11673, Saudi Arabia; d.aldailami@seu.edu.sa (D.A.A.); e.elsayed@seu.edu.sa (E.A.E.); 2Community Health Nursing Department, Faculty of Nursing, Damanhour University, Damanhour 22516, Egypt; 3Community Health Nursing Department, Faculty of Nursing, Zagazig University, Zagazig 44519, Egypt; mmdawood@nursing.zu.edu.eg; 4Community Health Nursing Department, Faculty of Nursing, Ain Shams University, Cario 11517, Egypt

**Keywords:** telehealth usability, adults, Kingdom of Saudi Arabia

## Abstract

**Background**: Due to technological advancements, the demand for easily accessible and convenient healthcare services is rising globally. Thus, telehealth is gaining momentum that was previously unheard of. The Kingdom of Saudi Arabia (KSA) actively embraces digital innovation in the healthcare industry through its ambitious Vision 2030 initiative. This study aimed to assess the perceived usability of telehealth for personalized healthcare among the KSA adult population in the post-COVID-19 era. **Methods**: This cross-sectional study used a convenience sample of 975 adults from the general population in KSA. A digital survey was used for data collection through Survey Monkey software. It contained two sections: personal and health data and the Telehealth Usability Questionnaire (TUQ). The data was collected over three months (July–September 2024) and investigated using the multinomial logistic regression analysis. **Results**: This study showed that 59.8% of the participants have initiated telehealth service use before the COVID-19 pandemic, and the most frequently used services are the issuance of sick leave (30.6%), tele-prescription (29.0%), virtual consultation (20.0%), tele-mental health services (19.4%), telemonitoring (18.6), and tele-referral (18.2%). A high total telehealth usability score was observed among 60.4% of the participants: 72.7% perceive telehealth as highly useful, 76.4% perceive it as easy to use, 60.1% have high satisfaction and intention for future use, 57.4% perceive high interface quality, and 51.8% perceive a high interaction quality. Conversely, 45.8% perceive low reliability of the telehealth system. Multinomial logistic regression showed that low education (AOR = 2.04; 95% CI = 1.16–4.85) and using virtual consultation (AOR = 0.98; 95% CI = 0.38–0.99) were predictors of low telehealth usability. However, being female (AOR = 1.67; 95% CI = 1.27–2.55), being in higher education (AOR = 1.47; 95% CI = 1.02–2.34), and living in the central KSA region (AOR = 1.37; 95% CI = 1.10–2.23) were predictors of high telehealth usability. Working status, income, and presence of chronic disease were not statistically significant predictors. **Conclusions**: Telehealth services have been highly usable in KSA even before the COVID-19 pandemic, with commonly used asynchronous services. Many social, technological, and system-related factors could affect the user experience and system reliability. Hence, telehealth developments are recommended to overcome such barriers, with future initiatives focusing on the flexibility and convenience of telehealth systems.

## 1. Introduction

There is a pressing need to provide efficient, convenient, and accessible healthcare services to meet the needs of the rising population in the Kingdom of Saudi Arabia (KSA), with a 2.1 yearly percentage change. Telehealth, with its potential to bridge the gap in healthcare services, is a crucial solution to this challenge. KSA ranks number 48 in the list of countries by population, and 92.3% of its population lives in urban areas with a median age of 29.6 years [1,2]. Like most developed nations, almost everyone has smartphones and internet access [3,4]. KSA heightened telehealth services even before the COVID-19 pandemic; these services date to 1990. In 2019, KSA issued novel regulations on telehealth to all clinical staff to provide them with an inclusive framework for telehealth use and support its implementation [5]. Telehealth is an umbrella concept for the use of information–communication technology (ICT) for clinical services (synchronous/real-time or asynchronous/store-and-forward/non-live) between patients and clinicians or between two clinicians physically distant from one another. In addition to other remote health-linked services, such as administration, continuing education, and provider training to support distance healthcare, information exchange, and access to healthcare services [6].

The KSA Ministry of Health (MOH) has recognized the significance of care digitization, mirrored the Saudi Vision 2030, constantly renewed its regulations concerning telehealth, and mandated its inclusion in governmental and private healthcare insurance coverage. This commitment to regulation is a substantial driver for the persistent adoption and utilization of telehealth, ensuring that it meets KSA’s highest standards of care. The scope of telehealth services in KSA covers various services and applications, including tele-consultation, tele-diagnosis, remote patient monitoring, tele-management, tele-surgery, tele-assistance, tele-expertise, tele-education, cross-border access, and emerging technology use [5,7,8].

The MOH draws the road for enhancing the accessibility of health services through designing numerous e-platforms for patients, population, and healthcare professionals, such as the MOH Formulary App, Sehhaty, Anat, outpatient virtual clinics, and 937 call centers initiated in 2013 and operating 24 h to serve the proportion of the population in emergency and routine care [7,8]. Throughout the COVID-19 pandemic, there has been a surge in telehealth applications to face the pandemic through screening suspected cases, providing long-term remote care, and tracking confirmed cases. These applications have generated several terminologies in the medical field, including “Seha/Health”, Tabaud/Distance, Tetamman/Rest assured, Mawid/Appointment, and Tawakklna/Empowering digital companion. The MOH has urged people to use mobile health applications instead of visiting primary care clinics during the pandemic. Thus, telehealth has been central to primary healthcare services for mitigating the spread of coronavirus [9,10]. The utilization of telehealth has marginally risen since the COVID-19 pandemic, but minute evidence exists about the backward change to the pre-pandemic levels [8,11].

Telehealth services have played an invaluable role in pandemic containment. In addition, across all healthcare institutions in KSA, telehealth efficiently managed expenses for patient healthcare services, including digital consultations, prescriptions, and follow-ups. These services are reachable with a finger tap, directing to a specific healthcare provider [12]. Telehealth applications have also monitored chronic illness progress and reduced unnecessary hospital visits. Thus, telehealth addresses geographic barriers and unequal access to specialized healthcare services [13,14].

Evidence shows that telehealth also facilitates the provision of personalized healthcare. The term “personomics” describes aspects of personalized healthcare based on understanding the patient’s health beliefs, personalities, social networks, economic resources, and other life circumstances that have noteworthy effects on how and when a specified health condition will be evident and how it will react to treatment [15]. Personalized healthcare is a predominant framework that provides an individualized plan to boost the patient’s health status and minimize disease. Using a standardized approach for unique patients can reduce success rates and result in costly and tiresome return visits. Personalized healthcare utilizes predictive technologies to identify individual health risks and facilitates patients’ engagement in their health to achieve the best health outcomes [16,17].

The telehealth delivery system must be usable for patients and healthcare professionals to realize its potential benefits. Usability indicates the extent to which system users can use a product to achieve their objectives effectively, efficiently, and satisfactorily in an identified use context. The usability concept is crucial in making digital systems more straightforward to learn and utilize [18]. Measuring telehealth technology usability proposes evaluating and enhancing the effectiveness of the technology used and services delivered. Usability measures the users’ experience, their confronting issues, and their satisfaction with various aspects of the technology or service. It also focuses on assessing the quality of the interactions between two remote sites (e.g., audiovisual and communication quality and ease of use) and the overall impression of the service (i.e., comfort level and satisfaction with the encounter) [19,20].

Despite these benefits, telehealth services face several implementation barriers, including users’ uncertainties about privacy and data security, unfamiliarity with the app or technology use, technical barriers, and infrastructural restrictions hindering telehealth services utilization [21,22,23]. In addition, there is the potential for users’ embarrassment or discomfort from using the camera [24]. Many studies have noticed that telehealth may be appropriate for modest problems that do not require physical examination and need more information in face-to-face consultations [25,26].

Evidence shows that the number of e-consultation users in KSA before COVID-19 in 2019 was 0.79 million, which increased during the pandemic in 2021 to 1.14 million, continues to grow to 1.30 million in 2024, and is expected to reach 1.41 million in 2029. Overall, the number of digital health services users before COVID-19 in 2019 was 9.25 million, which increased during the pandemic in 2021 to 11.01 million, then slightly decreased to 10.72 million in 2022, and now in 2024, it has reached 12.01 and is expected to reach 14.43 by 2029 [27]. However, the latest studies have pointed out that the KSA population still needs to gain awareness, knowledge, and experience with telehealth and is unwilling to use it due to trust issues [28,29]. A recent systematic review added that despite telehealth representing a major high-tech breakthrough in healthcare, its adoption has some drawbacks. Significant challenges are the limited availability of telehealth technologies and the focus on healthcare privatization, which imposes barriers toward adoption. In addition, healthcare personnel are not well-informed or conscious of the benefits linked to telehealth. Another significant barrier to the widespread adoption of telehealth is the substantial financial investment required to establish telemedicine and telehealth infrastructure [30]. Hence, there is a critical need to develop more effective telehealth apps and services. This could be achieved by investigating the population’s perceived benefits and usability of telehealth services while considering several demographic and technical factors influencing their use.

Therefore, the current study offers context for addressing the identified literature gap concerning telehealth usability. It aims to assess the perceived telehealth usability for personalized healthcare among adult populations. Using a cross-sectional design, the current study intends to capture information about the perceived usability pattern of telehealth services in the post-COVID-19 era among adults to take a snapshot of all KSA regions. This, in turn, helps in filling the literature gap, as most literature focuses on the pandemic era. Therefore, the main objectives of the current study are to explore the perceived usability pattern of telehealth services after COVID-19 compared to before and during the pandemic, identify the commonly used telehealth services among adults, and recognize the predictors of low and high telehealth services usability patterns. Thus, the current study further addresses and explores numerous barriers against telehealth services to facilitate the implementation of new strategies for better management. This aligns with the KSA healthcare sector transformation and the recommended digitization per the 2030 Saudi Vision. Hence, the findings will better explain why system users use or do not use telehealth. The findings can also draw the attention of several designers or vendors of telehealth systems to modify or adapt the system to improve the user experience.

## 2. Materials and Methods

### 2.1. Study Design: Population-Based Cross-Sectional Design

#### 2.1.1. Study Participants and Sampling

The current study’s target population was the general adult population living in KSA who fulfilled specific inclusion criteria, including those who were Saudi or non-Saudi, were older than 18, were men or women, used telehealth applications and services, and agreed to participate in this study. A convenience sample technique was employed to recruit the study sample through different social media platforms using Survey Monkey software (SurveyMonkey Inc., San Mateo, CA, USA, https://www.surveymonkey.com/) for data collection. To represent all KSA regions and lessen the selection bias, the researchers benefited from the properties of the Survey Monkey program to specify the inclusion criteria (all KSA regions, adult men and women, and ticking all the offered social media platforms, including Twitter, Instagram, LinkedIn, Facebook, and Snapchat). This meticulous approach ensured the study’s findings were comprehensive and reflected the KSA general adult population. Moreover, many rounds were made to ensure the survey was distributed to all KSA regions, further enhancing the study’s credibility.

The required sample size was determined using the following equation parameters for proportion in survey-type studies [31]: 95% confidence level (Z = 1.96 for alpha 0.05), margin of error or precision (E = 0.05), and the proportion of telehealth use in KSA (*p* = 56.2%), based on a recent study by Alshahrani et al., 2024 [32], while considering the sampling design effect (D = 2.5) for including five different KSA regions. The minimum calculated sample size was 947, and the final sample size was 975 participants.
$$N=\frac{Z_{\frac{\alpha }{2}}^{2}\times P\times \left(1-P\right)\times D}{E^{2}}$$

#### 2.1.2. Survey Development

The researchers designed a digital self-administered survey using credible and recent evidence to explore the perceived pattern of telehealth services usability after the COVID-19 pandemic. It comprised two parts:Personal and health data, such as age, gender, marital status, education, residence, income status, chronic disease type, time of initial use of telehealth service, and telehealth services commonly used.Telehealth Usability Questionnaire (TUQ) evaluates the user-perceived usability of telehealth services and applications, their implementation, and their effectiveness. It is a valid questionnaire designed by Parmanto, B. et al. in 2016, who proved all its attributes had good to excellent reliability (α ranged between 0.83–0.91). TUQ has 21 items distributed over six subscales: usefulness (3 items), ease of use (3 items), reliability (3 items), interface quality (4 items), interaction quality (4 items), and satisfaction and future use (4 items) (Parmanto, B. et al., 2016). It uses a five-point Likert scale from (1) “strongly disagree” to (5) “strongly agree”. The weighted mean scores were used to analyze the usability items, subscale scores, and TUQ scores: strongly disagree (1.00–1.80), disagree (1.81–2.60), neutral (2.61–3.40), agree (3.41–4.20), and strongly agree (4.21–5.00). It was further categorized into three levels: low (≤3.39), moderate (3.40–3.79), and high (≥3.80), to determine the level of perceived telehealth usability [33].

### 2.2. Survey Validity and Reliability

The investigators translated the TUQ into Arabic using the DeepL Translator software program version 24.11 (DeepL SE Co., Cologne, NW, Germany). An expert researcher translated the questionnaire back into English to guarantee its precision. A panel of five experts critically revised the instrument to investigate and agree upon its content and items’ wording, arrangement, and scoring. Then, the required adjustments were made based on panel feedback. The cut-off value of the Content Validity Index (CVI) was (>0.70), and the TUQ showed a satisfactory level (CVI = 0.89). The instrument’s internal consistency reliability was tested using Cronbach’s Alpha coefficient test (α) with a cut-off value of (>0.70). The overall TUQ proved satisfactory reliability (α = 0.82), and for its subscales (U = 0.85; EoU = 0.84; IFQ = 0.79; ITQ = 0.83; R = 0.80; SFU = 0.85). The instrument was piloted on 10% of the sample size (omitted from the overall study sample) to ensure the instrument’s clearness, simplicity, and wording. Accordingly, any noticed modifications were fixed.

### 2.3. Data Collection

Data was collected through an online self-administered questionnaire using Survey Monkey. The survey was piloted on 10% of the sample size (omitted from the primary sample) to guarantee its precision, wording, and pertinence, and the needed modifications were completed. It was made available through social media platforms, including Twitter, Instagram, LinkedIn, Facebook, and Snapchat. Screening questions for eligibility were set at the commencement of the survey concerning age and telehealth use, and if they were not relevant, the survey was terminated. It was open for three months (from July to September 2024). Many rounds of data collection were done to ensure that the survey reached all KSA regions. The registered filling-in time on the Survey Monkey Software was between 8 and 11 min, with a 93.5% response rate.

### 2.4. Statistical Analysis

Researchers deployed IBM Statistical software, version 27 (IBM Corp., Armonk, NY, USA). Descriptive statistics (mean scores with standard deviations with 95% confidence interval (CI)) were employed for continuous variables, while frequency and proportion were used for categorical variables. The normality of data was warranted using the Shapiro–Wilk test (*p* > 0.05). The analysis of variance test (one-way ANOVA) was used to test the significance of weighted mean scores differences between total TUQ and its domains based on participants’ personal and health data.

Multinomial logistic regression analysis examined low and high telehealth usability predictors while taking a moderate level as a reference. Researchers used the coefficient of determination of Cox and Snell R^2^ and Nagelkerke R^2^ values to evaluate model fitness, where higher values signify better model fit. Model significance was judged using the chi-squared test *p* value (*p* ≥ 0.05). Model goodness of fit was tested using the Hosmer and Lemeshow test, where a non-significant *p*-value (*p* > 0.05) signals model fitting for data [34]. A *p*-value less than 0.05 is considered statistically significant.

### 2.5. Ethical Consideration

The study obtained ethical approval from the Saudi Ministry of Health (IRB Log No. 24–106 E) and strictly followed the Declaration of Helsinki. The survey’s link thoroughly explained the study’s aim and provided the necessary elaboration for answering questions. The researchers obtained digital informed consent from each respondent before starting the survey to ensure voluntary participation. Respondents were informed of their right to decline their replies at any time and that their answers would be kept anonymous and solely used for the study.

## 3. Results

### 3.1. Participants’ Personal and Health Data

Table 1 illustrates that 35.8% of the studied participants are between 31 and 40 years old, with a mean age of 36.28 ± 10.05. More than half (59.7%) are females, 68.8% are married, and 91.4% are Saudis. Nearly two-thirds (68.1%) have a bachelor’s education, 62.4% are working, 32.1% and 20.6% live in the central or eastern regions, 41.6% have sufficient income, and 76.7% have no chronic diseases. More than half (59.8%) initiated using telehealth services before the COVID-19 pandemic. The frequently used telehealth services are the issuance of sick leave (30.6%), tele-prescription (29.0%), virtual consultation (20.0%), tele-mental health services (19.4%), telemonitoring (18.6), and tele-referral (18.2%).

### 3.2. TUQ Items and Domains Analysis by Participants’ Personal and Health Data

Table 2 shows the weighted mean of total usability scores (3.91 ± 0.690, 95% CI = 3.85–3.96) and the variations between its subscales and items. Domains of ease of use (4.14 ± 0.733, 95% CI = 4.10–4.18), usefulness (4.11 ± 0.749, 95% CI = 4.07–4.16), satisfaction and future use (3.93 ± 0.772, 95% CI = 3.88–3.98), and interface quality (3.92 ± 0.764, 95% CI = 3.87–3.96) have high weighted mean scores. However, domains of interaction quality (3.77 ± 0.841, 95% CI = 3.72–3.83) and reliability (3.58 ± 0.893, 95% CI = 3.52–3.63) have moderate weighted mean scores.

Table 3 shows that nearly two-thirds (60.4%) of the participants have a high telehealth usability score, while 21.3% have low usability, with slight variation between subscales. Around three-quarters of the participants perceive telehealth as highly useful (72.7%) and having ease of use (76.4%); 60.1% have high satisfaction and intention for future use. More than half (57.4%) perceive the telehealth system’s high interface quality, while 21.0% perceive its low quality. About half (51.8%) perceive telehealth experience as having a high interaction quality; however, 32.4% perceive it as low quality. Conversely, 45.8% of the participants perceive telehealth systems as having low reliability compared to 42.1% who perceive them as highly reliable.

Table 4 depicts statistically significant higher weighted mean differences between levels of the total TUQ by increasing age (F = 11.356, *p* = 0.042), being female (t = 5.436, *p* = 0.041), having higher education (F = 5.716, *p* = 0.034), and living in the eastern or central KSA region (F = 8.570, *p* = 0.044). Statistically significant higher weighted mean differences of the perceived telehealth usefulness are detected among older participants (F= 3.992, *p* = 0.044), females (t = 4.546, *p* = 0.033), and those having chronic diseases (t = 6.785, *p* = 0.021). Concerning the ease-of-use and interface-quality subscales, statistically significant higher weighted mean differences are noticed among highly educated participants (F = 4.206, *p* = 0.022), (F = 9.709, *p* = 0.041), respectively. Regarding the interaction quality subscale, statistically significant higher weighted mean differences are detected among younger participants (F = 6.829, *p* = 0.017), females (t = 3.123, *p* = 0.026), and those living in the eastern or central KSA region (F = 2.858, *p* = 0.028). The telehealth system reliability subscale statistically significantly differs by the type of commonly used telehealth service (F = 12.754, *p* = 0.006), where virtual consultation has low weighted mean scores (3.35 ± 0.905; 95% CI = 3.23–3.68). Finally, the satisfaction and future use subscale show statistically significant higher weighted mean differences among older participants (F = 2.326, *p* = 0.040), females (t = 6.568, *p* = 0.017), and those living in the eastern or central KSA region (F = 8.617, *p* = 0.000).

### 3.3. Multinomial Logistic Regression of the Predictors of Low and High Telehealth Usability

Table 5 illustrates that low educational status and virtual consultation are predictors of low telehealth usability among the studied participants. Lower educational level increases the probability of low telehealth usability by 2.04 times (AOR = 2.04; 95% CI = 1.16–4.85). Using virtual consultation reduces the likelihood of telehealth usability by 0.98 times (AOR = 0.98; 95% CI = 0.38, 0.99). On the other hand, being female, having higher education, and living in the central KSA region are predictors of high telehealth usability. Being female increases the probability of telehealth usability by 1.67 times (AOR = 1.67; 95% CI = 1.27–2.55). Increasing the educational level increases the likelihood of telehealth usability by 1.47 times (AOR = 1.47; 95% CI = 1.02–2.34). Finally, living in the central KSA region increases the probability of telehealth usability by 1.37 times (AOR = 1.37; 95% CI = 1.10–2.23).

## 4. Discussion

The present study found that nearly two-thirds of the participants rated telehealth as having high usability, and more than half had initiated telehealth use before the COVID-19 pandemic. These findings reflect the rising rate of telehealth service utilization among adults and the KSA healthcare system’s outstanding efforts in employing telehealth services in the pre-pandemic era that allowed more accessible adaptation during the health crisis and acting proactively for better management. Similarly, a community-based study in Jeddah by Albaghdadi and Al Daajani (2023) showed a positive perception, satisfaction, and approval that telehealth services significantly improved healthcare accessibility, where more than half of the participants started their use before the emergence of COVID-19 [24].

Concerning the frequently used telehealth services, our results align with local and global trends, where participants commonly used asynchronous services, whereas only 20% used synchronized virtual consultation services. Evidence has proven that asynchronous services have several advantages, including flexibility, convenience, ease of use, and compatibility with various medical conditions [24,35]. A recent study in Jeddah city by Albaghdadi and Al Daajani (2023) found that nearly three-quarters of the participants felt potential embarrassment or discomfort during synchronous services due to the presence of the camera and equipment [24]. In addition, many issues related to network or mobile apps are frequently reported barriers to telehealth services and are often associated with low utilization and satisfaction [36,37]. This finding matches the current study, which showed that synchronized virtual consultation services have the lowest weighted reliability mean scores and low interaction quality reported by about one-third of participants and moderate mean scores of the items comparing virtual consultation to in-person visits: “Using the telehealth system, I could see the clinician as if we met in person” and “I feel comfortable communicating with the clinician using the telehealth system”.

Our findings prove that telehealth usability significantly increases with age, being female, having higher education, and residency in the eastern or central KSA regions. This finding may be due to the urbanicity of these KSA areas, with central hospitals providing substantial health services and good internet connection facilitating telehealth services delivery. The study also found that lower educational status and virtual consultation were predictors of low telehealth usability. However, being female, having higher education, and living in the central KSA region are predictors of high telehealth usability. This finding illustrates the role of certain demographic variables in shaping the individual’s experience with telehealth services. It also highlights the need to focus on virtual consultations to increase population satisfaction and usability. Likewise, a cross-sectional study in Riyadh by Almalki et al. (2023) showed that female gender, higher education, and living in urban areas were associated with higher odds of telemedicine services use [38]. Al-Shroby et al. (2024) added that having a history of chronic disease was significantly associated with utilization and satisfaction with telehealth services [39]. However, a cross-sectional study during the COVID-19 pandemic by Aldhahir et al. (2022) explored the idea that telehealth utilization was relatively low, while using other applications that originated during the pandemic was high [36]. The authors attributed this to the need for more knowledge about telehealth services and their benefits and the focus on managing infection using specialized applications for COVID-19 cases. Along with the current study, they detected higher utilization patterns among females, adults, and highly educated individuals.

On the other hand, Almalki et al. (2023) showed that participants with low socioeconomic indexes and those older had lower odds of using telemedicine than those under thirty and those with high socioeconomic indexes [38]. This contradiction between Almalki’s study and the current one could be explained by the variance in the setting and the sample characteristics, where Almalki’s study was conducted among chronically ill patients in primary healthcare centers in the Riyadh region only. There were also age differences in the sample, with a high proportion of the elderly population also suffering from multiple comorbidities. However, the current study population’s mean age was 36.28 years, representing all KSA regions, and most had no chronic diseases.

Concerning the usability subscales variations, the present study shows that around three-quarters of the participants perceive telehealth as highly useful and easy to use, and about two-thirds have high satisfaction and future use intent. More than half have a high perception of the telehealth system’s interface quality and interaction quality of telehealth service. Conversely, nearly half perceive telehealth systems as having low reliability. These variations are mainly explained by several personal, demographic, and system-related factors that can affect the user experience, as the current study proves through an in-depth analysis of subscales. Education, gender, residency, and using virtual consultation services were the main predictors of the usability of telehealth services.

The current study shows that about three-quarters of participants perceive the high usefulness of telehealth services, which significantly increases with age, being female, and having chronically ill conditions. Such findings could be explained by the perceived improved health outcomes of various conditions, which can heighten their perceived amenability to telehealth, including the lower cost and accessibility of telehealth services [40]. Likewise, Record et al. (2021) explain the usefulness of telehealth in providing personalized healthcare through new communications technologies that enable more detailed descriptions of patients’ lives, which can be used to customize treatment and diagnosis plans [16]. A study in the United Arab Emirates by Abdool et al. (2021) also found high mean scores for telehealth usefulness among the participants [41]. However, Dawood and Alkadi (2022) found that more than two-thirds of participants disagreed with the usefulness of the Sehhaty application in KSA [42]. They reported that it did not accommodate their health needs or save time; thus, they preferred traditional visits. Contradiction in the perceived usefulness could be explained by cultural factors and social influence that moderate the relationship between perceived telehealth value and its acceptance and usability [43]. 

The current study depicts that over three-quarters of the participants perceive high ease of use of telehealth services, which is significantly high among highly educated participants. These findings highlight that educated people are more able to deal with modern technology, which is thus perceived as simple and easy to learn and to use for facilitating speedy work completion without deliberate efforts [30]. Similarly, two cross-sectional studies of the leading telehealth apps in KSA by Aldhahir et al. (2022) [36], who studied the “Seha” app, and Dawood and Alkadi (2022) [42], who studied the “Sehhaty” app, found that most participants agreed or strongly agreed that they were simple and easy to use. In addition, Albaghdadi and Al Daajani (2023) showed that most of their participants were young (26–45 years), indicating a higher propensity to use digital health tools, which was significantly associated with high satisfaction as younger cohorts are inclined to have higher digital literacy [24].

The current study finds a low proportion of perceived high telehealth system interface quality among participants (57.5%), while around one-fifth have either a low or moderate perception. The mean score of interface quality is significantly increasing among highly educated participants. This finding also highlights the role of education in dealing with technology. It draws attention to the need to upgrade the telehealth system’s functionality, capabilities, and navigation to make it more user-friendly and enhance the population’s interaction and experience. Abdel Nasser et al. (2021) showed that over half of the participants were satisfied with the system’s ease of registration and scheduling [44]. Conversely, a higher trend was demonstrated by Dawood and Alkadi (2022), who expressed that nearly two-thirds of the participants agreed with the telehealth system (Sehhatty) interface quality [42]. This contradiction could be explained by the focus of this conflicting study on one telehealth app, while the current study investigated a wide array of telehealth services (synchronous and asynchronous).

The present study finds that about half of the participants have a high perception of telehealth services interaction quality, and nearly one-third have a low perception. This perception significantly differs among younger participants, females, and those living in the eastern or central KSA regions. This could be explained by the ability of young people to deal efficiently with modern technologies and their abilities to express themselves virtually, like their modern virtual learning and work lives. In addition to fitting telehealth services as an option in overloaded females’ lives, the good internet connection available in any country’s urban areas improves the quality of virtual patient-provider interaction and system use. In accordance, Abdel Nasser et al. (2021) showed that the participants were satisfied with the quality of the telehealth service audio and visual image, understood the given recommendations, felt comfortable with the services, and could talk freely [44]. Despite the MOH efforts to accelerate the digital health revolution and empowerment by designing and sustaining high-quality telehealth services, technical problems associated with network or mobile apps are still a frequent barrier and are often associated with low utilization and satisfaction with telehealth services [36,37].

The current study detects that nearly half of the participants perceive telehealth systems as having low reliability compared to the more than two-fifths who perceive them as highly reliable. This relationship significantly differs by the commonly used telehealth service type, where virtual consultation has the lowest scores (Table 4). Such findings may be attributed to the nature of the virtual clinic and its dependency on the internet connection, which may vary between residential areas, in contrast to in-person consultations. Moreover, telehealth system reliability depends on how much guidance the system offers users in the event of an error to help them recover quickly and on the validity and reliability of data transmission [21,30]. Dawood and Alkadi (2022) showed a similar low-reliability trend where about one-third of the participants agreed or strongly agreed with telehealth system reliability [42]. Alshahrani et al. (2024) also found that traditional consultation preference among the KSA population was the most significant barrier against telehealth services [32].

Moreover, our findings show that nearly two-thirds of the participants are highly satisfied and intend to use telehealth in the future, which significantly increases by age, by being female, and among those living in the eastern or central KSA regions. Such findings could be partially explained by comorbidities among the adult population and specific issues that females may feel embarrassed about in in-person consultations that make telehealth the best fit for many females’ health issues. Residential areas may also affect internet connection and interaction with healthcare providers, affecting participants’ satisfaction [29,44]. Likewise, Aldhahir et al. (2022) showed that most participants were satisfied with the telehealth services (Seha app) and agreed that the answers delivered by the app were accurate and linked to their conditions [36]. AlBaghdadi and Al Daajani (2023) also found high satisfaction with telehealth services, especially in specific domains, including accessible communication and ease of use. They also showed that older age predicted poor satisfaction [24]. The difference in age group could explain this contradiction: our study included mainly the adult population, not only the older ones.

To sum up, the present study explored the perceived high usability of telehealth services among adults in KSA in the post-pandemic era compared to during or before the pandemic. It found that asynchronous services were mainly emphasized by adults due to concerns about telehealth system reliability and interface quality, as well as the quality of interaction during synchronized services. The study also explored several personal and demographic factors influencing user experience and satisfaction. Thus, these current challenges in telehealth implementation demand urgent attention. The cultural and social influence of preferring traditional face-to-face consultation hinders the adoption of new technology, especially the synchronous option. The technological barriers related to the telehealth system and network negatively impact the user experience and system reliability. Moreover, virtual consultation stability and security are paramount to guaranteeing the patient’s privacy and promoting trust [19,21,45].

### Study Strengths, Limitations, and Future Implications

This study focuses on the usability of telehealth services in the post-pandemic era. It compares it to the usability pattern before and during the pandemic, whereas most literature focuses on the surge of telehealth services and applications during the pandemic. It explores the commonly used telehealth services among adults in KSA in the post-pandemic era. The study also used a standardized TUQ with high validity and reliability using a holistic view of usability construct through six subscales. In addition, it is a nationwide study that uses a large sample size with a nearly proportional percentage from each KSA region, which captures a holistic image of telehealth usability. However, various limitations arise linked to the online self-reported data and the lack of sample randomization; however, the study tried to overcome this by considering the sampling design effect to reduce the sample variance and error. In addition to reaching all KSA geographical regions, which provided variations between regions according to the degree of urbanicity of the region, this heterogeneity highlights the need for distinct rural/urban analysis in further investigation. Another limitation is linked to the inherent nature of the cross-sectional studies of the limited causality inferences.

## 5. Conclusions

The current study concludes that telehealth usability is high among nearly two-thirds of the KSA population, with more than half initiating telehealth use before the COVID-19 pandemic. The predominant type of telehealth services used is asynchronous services. The usability scores significantly increase with age, being female, having higher education, and residency in the eastern or central KSA regions. This study also depicts that low educational status and virtual consultation are predictors of low telehealth usability. However, being female, having higher education, and living in the central KSA region are predictors of high telehealth usability. Moreover, most participants perceive high telehealth usefulness, ease of use, satisfaction, and future use intent. A low perception of the telehealth system’s interface and interaction quality is observed, with a prevalent low telehealth system reliability observed among nearly half of the participants.

Hence, the researchers recommend that future telehealth developments address social, cultural, and technological barriers. Future initiatives should focus on the flexibility and convenience of telehealth systems to resonate with the expectations of different demographic groups, notably rural adults. Public health education and awareness campaigns targeting telehealth benefits and use through informational brochures or manuals are recommended. Moreover, action research projects to expand telehealth services based on consumer feedback and shareholder engagement in KSA are warranted.

## Figures and Tables

**Table 1 healthcare-13-00062-t001:** Distribution of the participants according to personal and health data (n = 975).

Parameters	No.	%
Age (in years)		
20–30	331	33.9
31–40	349	35.8
41–50	211	21.6
51–60	69	7.1
>60	15	1.5
Mean ± SD (95% CI)	36.28 ± 10.05 (35.65–36.91)	
(Min–Max)	(20–63)	
Gender		
Males	393	40.3
Females	582	59.7
Marital status		
Married	671	68.8
Not married	304	31.2
Nationality		
Saudi	891	91.4
Non-Saudi	84	8.6
Educational level		
Primary education	36	3.7
High school	141	14.5
Bachelor education	664	68.1
Postgraduate education	134	13.7
Working status		
Working	610	62.6
Not working	335	37.4
Residence region		
Eastern	201	20.6
Western	174	17.8
Central	313	32.1
Southern	159	16.3
Northern	128	13.1
Income status		
Insufficient	311	31.9
Sufficient	406	41.6
Sufficient and saving	258	26.5
Chronic disease		
No	748	76.7
Yes	227	23.3
Diabetes	81	35.7
Hypertension	65	28.6
Asthma	29	12.8
Anemia	27	11.9
Heart diseases	15	6.6
Thyroid diseases	10	4.4
Initial use of telehealth services		
Never used	10	1.1
Before COVID-19	583	59.8
During COVID-19	382	39.2
Commonly used telehealth service ^#^		
Issuance of sick leave	298	30.6
Tele-mental health services	189	19.4
Tele-prescription	283	29.0
Virtual consultation	197	20.2
Tele-pathology and tele-radiology	150	15.4
Tele-referral	177	18.2
Tele-monitoring	181	18.6

SD = standard deviation; ^#^ Items not mutually exclusive.

**Table 2 healthcare-13-00062-t002:** Distribution of the TUQ by item and subscales (n = 975).

Domains	Mean	SD	Strongly Disagree n (%)	Disagree n (%)	Neutral n (%)	Agree n (%)	Strongly Agree n (%)
Usefulness (U) (items 1–3)							
1. Telehealth improves my access to healthcare services.	4.14	0.867	11 (1.1)	34 (3.2)	138 (14.1)	412 (42.3)	380 (39.0)
2. Telehealth saves me time traveling to a hospital or specialist clinic.	4.16	0.828	3 (0.3)	43 (4.4)	122 (12.5)	435 (44.6)	372 (38.2)
3. Telehealth provides for my healthcare needs.	4.04	0.881	5 (0.5)	50 (5.1)	181 (18.6)	407 (41.7)	332 (34.1)
Weighted mean ± SD (95% CI)	4.11 ± 0.749 (4.07–4.16)						
Ease of Use (EoU) (items 4–6)							
4. It was simple to use this system.	4.16	0.837	12 (1.2)	25 (2.6)	128 (13.1)	438 (44.9)	372 (38.2)
5. It was easy to learn to use the system.	4.23	0.809	6 (0.6)	31 (3.2)	104 (10.7)	428 (43.9)	406 (41.6)
6. I believe I could become productive quickly using this system.	4.04	0.923	12 (1.2)	53 (5.4)	167 (17.1)	399 (40.9)	344 (35.3)
Weighted mean ± SD (95% CI)	4.14 ± 0.733 (4.10–4.18)						
Interface Quality (IFQ) (items 7–10)							
7. The way I interact with this system is pleasant.	3.91	0.893	8 (0.8)	58 (5.9)	213 (21.8)	429 (44.0)	267 (27.4)
8. I like using the system.	3.94	0.923	14 (1.4)	57 (5.8)	192 (19.7)	425 (43.6)	287 (29.4)
9. The system is simple and easy to understand.	4.15	0.849	9 (0.9)	39 (4.0)	119 (12.2)	442 (45.3)	366 (37.5)
10. This system can do everything I would want it to be able to do.	3.67	1.004	22 (2.3)	102 (10.5)	265 (27.2)	371 (38.1)	215 (22.1)
Weighted mean ± SD (95% CI)	3.92 ± 0.764 (3.87–3.96)						
Interaction Quality (ITQ) (items 11–14)							
11. I could easily talk to the clinician using the telehealth system.	3.88	0.953	17 (1.7)	63 (6.5)	216 (22.2)	401 (41.1)	278 (28.5)
12. I could hear the clinician using the telehealth system.	3.84	0.924	9 (0.9)	65 (6.7)	257 (26.4)	386 (39.6)	258 (26.5)
13. I felt I was able to express myself effectively.	3.80	0.952	8 (0.8)	93 (9.5)	233 (23.9)	395 (40.5)	246 (25.2)
14. Using the telehealth system, I could see the clinician as if we met in person.	3.58	1.006	22 (2.3)	112 (11.5)	319 (32.7)	326 (33.4)	196 (20.1)
Weighted mean ± SD (95% CI)	3.77 ± 0.841 (3.72–3.83)						
Reliability (R) (items 15–17)							
15. I think the visits provided over the telehealth system are the same as in-person visits.	3.47	1.08	39 (4.0)	145 (14.9)	287 (29.4)	324 (33.2)	180 (18.5)
16. Whenever I made a mistake using the system, I could quickly recover.	3.68	0.978	21 (2.2)	87 (8.9)	283 (29.0)	374 (38.4)	210 (21.5)
17. The system gave error messages that told me how to fix problems.	3.59	1.012	38 (3.9)	78 (8.0)	322 (33.0)	349 (35.8)	188 (19.3)
Weighted mean ± SD (95% CI)	3.58 ± 0.893 (3.52–3.63)						
Satisfaction and Future Use (SFU) (items 18–21)							
18. I feel comfortable communicating with the clinician using the telehealth system.	3.78	0.991	17 (1.7)	97 (9.9)	218 (22.4)	397 (40.7)	246 (25.2)
19. Telehealth is an acceptable way to receive healthcare services.	3.99	0.873	10 (1.0)	48 (4.9)	170 (17.4)	457 (46.9)	290 (29.7)
20. I would use telehealth services again.	3.88	0.932	16 (1.6)	59 (6.1)	215 (22.1)	420 (43.1)	265 (27.2)
21. Overall, I am satisfied with this telehealth system.	4.08	0.862	16 (1.6)	33 (3.4)	133 (13.6)	472 (48.4)	321 (32.9)
Weighted mean ± SD (95% CI)	3.93 ± 0.772 (3.88, 3.98)						
Total weighted mean ± SD (95% CI) = 3.91± 0.690 (3.85–3.96)							

SD = standard deviation, CI = confidence interval. Total weighted mean (items 1 to 21).

**Table 3 healthcare-13-00062-t003:** Distribution of the total weighted mean score levels of the TUQ and its subscales (n = 975).

Domains	Low n(%)	Moderate n(%)	High n(%)
Usefulness	177 (18.2)	99 (9.1)	709 (72.7)
Ease of Use	174 (17.8)	56 (5.7)	745 (76.4)
Interface Quality	205 (21.0)	210 (21.5)	560 (57.4)
Interaction Quality	316 (32.4)	154 (15.8)	505 (51.8)
Reliability	447 (45.8)	118 (12.1)	410 (42.1)
Satisfaction and Future Use	213 (21.8)	176 (18.1)	586 (60.1)
Total TUQ score	208 (21.3)	178 (18.3)	589 (60.4)

**Table 4 healthcare-13-00062-t004:** Analysis of weighted mean differences of the perceived TUQ domains by the participants’ personal and health data (N = 975).

Parameters	Total	U	EoU	IFQ	ITQ	R	SFU
	x¯ ± SD (95% CI)						
Age (in years)							
- 20–30	3.82 ± 0.694 (3.84–3.99)	3.77 ± 0.816 (3.69–4.02)	4.19 ± 0.723 (4.11–4.26)	3.95 ± 0.741 (3.87–4.03)	3.80 ± 0.824 (3.70–3.88)	3.64 ± 0.886 (3.54–3.73)	3.99 ± 0.717 (3.91–4.06)
- 31–40	3.78 ± 0.765 (3.72–3.93)	3.79 ± 0.815 (3.76–4.16)	4.16 ± 0.716 (4.09–4.24)	3.93 ± 0.760 (3.85–4.01)	3.81 ± 0.865 (3.71–3.89)	3.57 ± 0.880 (3.47–3.66)	3.93 ± 0.781 (3.84–4.01)
- 41–50	3.97 ± 0.566 (3.84–4.11)	4.15 ± 0.737 (4.06–4.23)	4.17 ± 0.646 (4.02–4.33)	3.84 ± 0.822 (3.73–3.95)	3.84 ± 0.837 (3.30–4.23)	3.52 ± 0.918 (3.40–3.65)	3.83 ± 0.864 (3.71–3.95)
- 51–60	3.95 ± 0.653 (3.88–4.02)	4.27 ± 0.589 (4.13–4.41)	4.04 ± 0.788 (3.93–4.14)	3.94 ± 0.680 (3.78–4.10)	3.76 ± 0.705 (3.59–3.93)	3.62 ± 0.882 (3.41–3.84)	4.06 ± 0.635 (3.91–4.21)
- >60	3.75 ± 0.793 (3.34–4.22)	4.11 ± 0.741 (4.03–4.18)	4.00 ± 0.845 (3.53–4.47)	3.80 ± 0.892 (3.31–4.29)	3.71 ± 0.874 (3.59–3.82)	3.31 ± 1.035 (2.74–3.88)	3.65 ± 0.844 (3.18–4.09)
Sig.	F = 11.356, *p* = 0.042 *	F = 3.992, *p* = 0.044 *	F = 1.650, *p* = 0.816	F = 0.829, *p* = 0.507	F = 6.829, *p* = 0.017 *	F = 0.962, *p* = 0.428	F = 2.326, *p* = 0.040 *
Gender							
- Males	3.78 ± 0.694 (3.80–3.94)	3.79 ± 0.772 (3.75–4.10)	4.09 ± 0.754 (4.02–4.17)	3.87 ± 0.780 (3.80–3.95)	3.54 ± 0.820 (3.36–3.82)	3.54 ± 0.885 (3.45–3.62)	3.78 ± 0.766 (3.74–3.96)
- Females	3.94 ± 0.688 (3.88–3.99)	4.28 ± 0.735 (4.07–4.20)	4.17 ± 0.718 (4.11–4.23)	3.95 ± 0.718 (3.88–4.00)	3.87 ± 0.856 (3.73–3.89)	3.61 ± 0.898 (3.54–3.68)	3.96 ± 0.776 (3.90–4.03)
Sig.	t = 5.436, *p* = 0.041 *	t = 4.546, *p* = 0.033 *	t = 1.553, *p* = 0.121	t = 1.432, *p* = 0.159	t = 3.123, *p* = 0.026 *	t = 1.293, *p* = 0.196	t = 6.568, *p* = 0.017 *
Marital status							
- Married	3.90 ± 0.683 (3.86–3.96)	4.11 ± 0.742 (4.05–4.17)	4.13 ± 0.732 (4.08–4.19)	3.92 ± 0.759 (3.87–3.97)	3.77 ± 0.842 (3.71–3.83)	3.58 ± 0.891 (3.51–3.63)	3.93 ± 0.769 (3.97–3.99)
- Not married	3.91 ± 0.703 (3.84–3.99)	4.12 ± 0.767 (4.03–4.21)	4.16 ± 0.735 (4.08–4.24)	3.91 ± 0.775 (3.83–3.99)	3.79 ± 0.841 (3.69–3.88)	3.57 ± 0.909 (3.50–3.90)	3.94 ± 0.784 (3.85–4.02)
Sig.	t = 0.163, *p* = 0.870	t = 0.230, *p* = 0.818	t = 0.616, *p* = 0.538	t = 0.151, *p* = 0.880	t = 0.281, *p* = 0.779	t = 1.270, *p* = 0.204	t = 0.126, *p* = 0.900
Nationality							
- Saudi	3.90 ± 0.688 (3.85–3.94)	4.10 ± 0.753 (4.05–4.15)	4.14 ± 0.728 (4.08–4.19)	3.89 ± 0.765 (3.82–4.17)	4.78 ± 0.838 (3.72–3.83)	3.57 ± 0.878 (3.52–3.65)	3.92 ± 0.770 (3.87–3.97)
- Non-Saudi	3.99 ± 0.718 (3.84–4.14)	4.24 ± 0.700 (4.09–4.38)	4.20 ± 0.792 (4.03–4.37)	3.69 ± 0.747 (3.64–3.96)	4.75 ± 0.879 (3.56–3.94)	3.70 ± 0.927 (3.47–3.68)	4.05 ± 0.799 (3.88–4.22)
Sig.	t = 1.143, *p* = 0.253	t = 1.646, *p* = 0.436	t = 0.791, *p* = 0.429	t = 1.121, *p* = 0.836	t = 0.274, *p* = 0.784	t = 0.184, *p* = 0.854	t = 1.508, *p* = 0.132
Educational level							
- Primary	3.79 ± 0.697 (3.74–3.95)	4.09 ± 0.765 (4.03–4.15)	3.76 ± 0.779 (3.72–4.00)	3.69 ± 0.855 (3.50–4.28)	3.95 ± 0.849 (3.66–4.24)	3.71 ± 0.917 (3.39–4.02)	4.01 ± 0.793 (3.74–4.27)
- High school	3.77 ± 0.658 (3.75–4.01)	4.17 ± 0.702 (4.05–4.28)	4.10 ± 0.737 (4.04–4.24)	3.99 ± 0.757 (3.87–4.11)	3.86 ± 0.825 (3.72–3.99)	3.62 ± 0.897 (3.39–4.02)	4.02 ± 0.785 (3.89–4.14)
- Bachelor	4.00 ± 0.746 (3.75–4.24)	4.29 ± 0.699 (4.04–4.52)	4.14 ± 0.732 (4.09–4.20)	3.90 ± 0.767 (3.84–3.96)	3.75 ± 0.855 (3.68–3.81)	3.57 ± 0.899 (3.50–3.64)	3.90 ± 0.773 (3.84–3.96)
- Postgraduate	3.97 ± 0.680 (3.85–4.08)	4.12 ± 0.733 (3.99–4.24)	4.14 ± 0.729 (4.02–4.26)	3.87 ± 0.734 (3.76–4.01)	3.78 ± 0.787 (3.64–3.91)	3.53 ± 0.855 (3.39–3.68)	3.97 ± 0.779 (3.84–4.10)
Sig.	F = 5.716, *p* = 0.034 *	F = 0.075, *p* = 0.818	F = 4.206, *p* = 0.022 *	F = 9.709, *p* = 0.041 *	F = 1.272, *p* = 0.283	F = 0.974, *p* = 0.224	F = 1.181, *p* = 0.316
Working status							
- Working	3.93 ± 00.698 (3.87–3.98)	4.13 ± 0.765 (4.03–4.19)	4.16 ± 0.737 (4.09–4.21)	3.93 ± 0.770 (3.87–3.99)	3.80 ± 0.833 (3.74–3.87)	3.60 ± 0.898 (3.53–3.68)	3.93 ± 0.788 (3.87–3.99)
- Not working	3.88 ± 00.678 (3.81–3.95)	4.09 ± 0.723 (4.01–4.13)	4.12 ± 0.727 (4.04–4.19)	3.90 ± 0.755 (3.82–3.97)	3.72 ± 0.854 (3.63–3.81)	3.54 ± 0.885 (3.45–3.63)	3.93 ± 0.748 (3.86–4.00)
Sig.	t = 0.914, *p* = 0.361	t = 0.773, *p* = 0.440	t = 0.763, *p* = 0.446	t = 0.592, *p* = 0.554	t = 1.498, *p* = 0.134	t = 0.069, *p* = 0.285	t = 0.026, *p* = 0.979
Residence Region							
- Eastern	3.94 ± 0.731 (3.81–4.06)	4.14 ± 0.774 (4.05–4.23)	4.17 ± 0.715 (4.05–4.28)	3.91 ± 0.813 (3.80–4.02)	3.85 ± 0.939 (3.72–3.99)	3.53 ± 0.953 (3.41–3.65)	3.95 ± 0.829 (3.83–4.06)
- Western	3.89 ± 0.744 (3.78–3.99)	4.11 ± 0.729 (3.99–4.22)	4.14 ± 0.761 (4.01–4.28)	3.85 ± 0.706 (3.75–3.96)	3.69 ± 0.796 (3.57–3.81)	3.54 ± 0.809 (3.43–3.67)	3.93 ± 0.729 (3.80–4.02)
- Central	3.95 ± 0.694 (3.87–4.02)	4.17 ± 0.766 (4.03–4.30)	4.20 ± 0.736 (4.12–4.28)	3.96 ± 0.768 (3.88–4.05)	3.94 ± 0.816 (3.79–3.98)	3.60 ± 0.922 (3.50–3.70)	3.94 ± 0.787 (3.85–4.02)
- Southern	3.78 ± 0.641 (3.73–4.00)	4.08 ± 0.763 (3.97–4.18)	4.10 ± 0.758 (3.99–4.21)	3.89 ± 0.716 (3.78–4.00)	3.77 ± 0.840 (3.64–3.90)	3.57 ± 0.826 (3.44–3.69)	3.79 ± 0.715 (3.72–4.02)
- Northern	3.76 ± 0.634 (3.71–3.95)	4.07 ± 0.691 (3.97–4.17)	4.06 ± 0.689 (3.96–4.17)	3.93 ± 0.811 (3.79–4.07)	3.77 ± 0.805 (3.63–3.91)	3.66 ± 0.917 (3.51–3.83)	3.78 ± 0.782 (3.70–4.01)
Sig.	F = 8.570, *p* = 0.044 *	F = 0.517, *p* = 0.524	F = 1.219, *p* = 0.713	F = 0.626, *p* = 0.644	F = 2.858, *p* = 0.028 *	F = 0.562, *p* = 0.690	F = 8.617, *p* = 0.000 *
Income Status							
- Insufficient	3.89 ± 0.687 (3.81–3.97)	4.07 ± 0.754 (3.99–4.15)	4.12 ± 0.735 (4.04–4.20)	3.90 ± 0.759 (3.82–3.99)	3.77 ± 0.852 (3.68–3.87)	3.56 ± 0.891 (3.46–3.66)	3.91 ± 0.759 (3.82–3.99)
- Sufficient	3.95 ± 0.699 (3.88–4.02)	4.13 ± 0.746 (4.06–4.21)	4.17 ± 0.731 (4.10–4.25)	3.95 ± 0.777 (3.83–4.03)	3.81 ± 0.843 (3.73–3.89)	3.63 ± 0.893 (3.55–3.72)	3.98 ± 0.788 (3.91–4.06)
- Sufficient and saving	3.88 ± 0.681 (3.76–3.96)	4.13 ± 0.749 (4.04–4.23)	4.13 ± 0.734 (4.03,4.21)	3.88 ± 0.749 (3.79–3.97)	3.72 ± 0.823 (3.61–3.82)	3.52 ± 0.894 (3.41–3.63)	3.88 ± 0.764 (3.79–3.98)
Sig.	F = 1.087, *p* = 0.338	F = 0.743, *p* = 0.476	F = 0.671, *p* = 0.512	F = 0.851, *p* = 0.427	F = 1.002, *p* = 0.367	F = 1.395, *p* = 0.248	F = 1.548, *p* = 0.213
Chronic disease							
- No	3.82 ± 0.748 (3.78–3.99)	3.78 ± 0.731 (3.67–4.11)	4.16 ± 0.716 (4.10–4.21)	3.89 ± 0.823 (3.78–3.99)	3.77 ± 0.849 (3.67–3.87)	3.56 ± 0.949 (3.43–3.68)	3.89 ± 0.838 (3.78–3.99)
- Yes	3.92 ± 0.672 (3.87–3.97)	4.23 ± 0.810 (4.07–4.17)	4.09 ± 0.789 (3.99–4.19)	3.93 ± 0.745 (3.87–3.93)	3.78 ± 0.840 (3.7–3.83)	3.59 ± 0.876 (3.52–3.65)	3.94 ± 0.752 (3.89–3.99)
Sig.	t = 0.738, *p* = 0.441	t = 6.785, *p* = 0.021 *	t = 1.193, *p* = 0.233	t = 0.653, *p* = 0.514	t = 0.070, *p* = 0.944	t = 0.483, *p* = 0.629	t = 0.939, *p* = 0.348
Commonly used telehealth service							
- Issuance of sick leave	3.89 ± 0.677 (3.80–3.98)	4.062 ± 0.724 (3.96–4.16)	4.11 ± 0.757 (4.00–4.21)	3.92 ± 0.740 (3.82–4.03)	3.80 ± 0.800 (3.68–3.91)	3.51 ± 0.863 (3.39–3.63)	3.94 ± 0.736 (3.84–4.05)
- Tele-mental health services	3.89 ± 0.681 (3.78–4.02)	4.12 ± 0.753 (3.99–4.25)	4.09 ± 0.739 (3.97–4.22)	3.89 ± 0.755 (3.75–4.02)	3.74 ± 0.849 (3.59–3.89)	3.65 ± 0.823 (3.51–3.79)	3.91 ± 0.750 (3.78–4.04)
- Tele-prescription	3.88 ± 0.697 (3.79–3.97)	4.13 ± 0.748 (4.04–4.22)	4.12 ± 0.757 (4.02–4.22)	3.89 ± 0.777 (3.80–3.99)	3.70 ± 0.833 (3.59–3.81)	3.53 ± 0.881 (3.43–3.65)	3.88 ± 0.810 (3.78–3.98)
- Virtual consultation	3.90 ± 0.642 (3.80–4.00)	4.09 ± 0.704 (3.98–4.21)	4.17 ± 0.647 (4.07–4.28)	3.91 ± 0.713 (3.79–4.02)	3.79 ± 0.819 (3.65–3.92)	3.35 ± 0.905 (3.23–3.68)	3.93 ± 0.731 (3.82–4.06)
- Tele-pathology and tele-radiology	3.95 ± 0.705 (3.81–4.09)	4.16 ± 0.742 (4.02–4.31)	4.15 ± 0.738 (4.00–4.30)	3.92 ± 0.796 (3.76–4.07)	3.85 ± 0.856 (3.67–4.02)	3.64 ± 0.914 (3.46–3.82)	3.96 ± 0.786 (3.81–4.12)
- Tele-referral	3.84 ± 0.756 (3.67–4.01)	4.02 ± 0.896 (3.82–4.22)	4.09 ± 0.825 (3.91–4.28)	3.88 ± 0.841 (3.69–4.07)	3.65 ± 0.896 (3.45–3.87)	3.52 ± 0.987 (3.30–3.75)	3.87 ± 0.839 (3.69–4.07)
- Tele-monitoring	4.09 ± 0.713 (3.94–4.25)	4.25 ± 0.751 (4.08–4.41)	4.33 ± 0.625 (4.20–4.47)	4.07 ± 0.779 (3.90–4.25)	3.97 ± 0.905 (3.77–9.17)	3.84 ± 0.942 (3.63–4.05)	4.10 ± 0.776 (3.93–4.27)
Sig.	F = 1.258, *p* = 0.274	F = 0.901, *p* = 0.494	F = 1.195, *p* = 0.307	F = 0.686, *p* = 0.661	F = 1.507, *p* = 0.173	F = 12.754, *p* = 0.006 *	F = 0.949, *p* = 0.459

x¯ = weighted mean; SD = standard deviation; CI: confidence interval; F = ANOVA test (one-way analysis of variance); independent *t*-test; * significant at <0.05.

**Table 5 healthcare-13-00062-t005:** Multinomial logistic regression model of low and high telehealth usability predictors.

Predictors	Low	High
	AOR (95% CI)	AOR (95% CI)
Age (in years)		
20–30	0.56 (0.08–3.75)	0.53 (0.11–2.66)
31–40	0.45 (0.07–3.01)	0.46 (0.09–2.28)
41–50	0.33 (0.04–2.37)	0.67 (0.14–3.32)
51–60	0.17(0.11–4.22)	0.48 (0.09–2.48)
>60	Ref	
Gender		
Females	1.36 (0.95–1.94)	1.67 (1.27–2.55) *
Males	Ref	
Marital status		
Married	0.98 (0.64–1.56)	1.25 (0.86–1.82)
Not married	Ref	
Nationality		
Saudi	0.88 (0.39–1.94)	0.67 (0.35–1.27)
Non-Saudi	Ref	
Educational level		
Primary education	2.04 (1.61–4.85) *	1.61 (0.59–4.41)
High school	1.44 (0.66–3.16)	1.63 (0.88–3.03)
Bachelor education	1.75 (0.97–3.17)	1.47 (1.02–2.34) *
Postgraduate education	Ref	
Working status		
Working	1.30 (0.85–1.99)	1.31 (0.91–1.86)
Not working	Ref	
Income status		
Insufficient	1.15 (0.68–1.96)	1.13 (0.73–1.75)
Sufficient	1.18 (0.71–1.96)	1.24 (0.82–1.89)
Sufficient and saving	Ref	
Residence region		
Eastern	1.57 (0.73–3.38)	1.67 (0.88–3.17)
Western	1.64 (0.86–3.15)	1.20 (0.69–2.08)
Central	1.17 (0.64–2.13)	1.37 (1.10–2.23) *
Southern	0.94 (0.64–1.84)	0.89 (0.51–1.53)
Northern	Ref	
Chronic disease		
Yes	0.92 (0.51–1.65)	0.87 (0.52–1.43)
No	Ref	
Commonly used telehealth service		
Issuance of sick leave	1.15 (0.47–2.83)	0.74 (0.36–1.51)
Tele-mental health services	1.86 (0.71–4.87)	1.05 (0.47–2.33)
Tele-prescription	1.30 (0.54–3.14)	0.83 (0.41–1.67)
Virtual consultation	0.98 (0.38–0.99) *	0.67 (0.32–1.41)
Tele-pathology and tele-radiology	1.39 (0.49–3.92)	1.04 (0.45–2.41)
Tele-referral	1.11 (0.39–3.14)	0.64 (0.28–1.48)
Tele-monitoring	Ref	

AOR: adjusted odds ratio; CI: confidence interval; * significant at *p* < 0.05.

## Data Availability

Data can be obtained from the corresponding author.

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
