# Peer review of "Perceived Telehealth Usability for Personalized Healthcare Among the Adult Population in Saudi Arabia: A Cross-Sectional Study in the Post-COVID-19 Era"

_healthcare, 2025, doi:10.3390/healthcare13010062_

Round 1

Reviewer 1 Report

Comments and Suggestions for Authors

In this study, the authors aimed to assess the usability of Telehealth for Personalized Healthcare among the adult population in Saudi Arabia after the Covid-19 period. It is clear that there has been a global increase in interest in telehealth recently, which has introduced a new service model in the healthcare sector and could transform the field in the future. While this study is significant in this context, there are several issues that need further clarification:

  1. The data collection took place during the summer (July-September), which may introduce a seasonal bias. An explanation should be provided to address this potential effect.

  2. The final paragraph of the Introduction should offer a more detailed summary of the study, including the research objectives, the methodology employed, and the type of data used.

  3. A brief paragraph outlining the structure of the study should be added at the end of the Introduction.

  4. Is it expected that the rising trend in telehealth use will persist in the future, similar to the post-pandemic period?

  5. The section on limitations and future research should be expanded. The current discussion lacks depth.

Comments on the Quality of English Language

 The English could be improved to more clearly express the research.

Author Response

Dear Reviewer 1,

We thank you for your valuable comments and suggestions for improvements.

All the points you raised are addressed (as in the attached file).

English editing was done (the report was attached, done by Grammarly professional premium account of the corresponding author) along with the modified version of the manuscript. 

Regards

Reviewer 2 Report

Comments and Suggestions for Authors

General Comments to Authors:

This piece covers the usability of telehealth for personalized healthcare in the diversity of regions in Saudi Arabia. The study is cross-sectional, lacking a pre/post design. Studies in this health-related area tend to be cross-sectional, and the authors have already noted this point in their limitations subsection. In their Discussion the authors have cited quite a number of corroborative studies. The paper should indicate in its beginning and Conclusions what it additionally contributes to the body of telehealth literature given the variety of other similar studies. Paper requires a considerable amount of explanatory fill in. The need for grammar correction in this piece is middle of the road.

Specific Comments to Authors:

P. 1, Para. 1:

line 19: Please specify whether the adults are from the general population, a hospital-based population, or the community of professionals

line 33: Please mention those variables that do not serve as statistically useful predictors

P. 3, Para. 2, line 104:

Please list the absolute #, if available, in addition to the percentage

P. 3, Para. 5, line 139:

State how participants were identified. Was a directory or registry of some kind used?

P. 4, Para. 6, line 197:

Is the Confidence Interval a 95% Confidence Interval? Please specify

P. 5, Para. 2, lines 208, 9:

Doe the authors mean a P-value of 0.5 or 0.05?

P. 6, Para. 1, line 235:

The highest weighted mean scores  ->  The highest weighted mean scores on a scale from X to Y   [Please specify]

P. 8, bottom of Table 2, line 241:

Include scale range (from X to Y) in the table legend

P. 8, Para. 2, line 254 and P. 10, Table 4:

Authors need to double-check significance level (P=0.047) of overall age mean difference given that the distribution is shaped like a normal curve which is decreasing, not rising at the end

P. 12, Table 2, Telehealth service as a parameter:

Authors should include in their Discussion 1 or 2 reasons why the analysis of the weighted mean differences of the perceived Telehealth service scores, 7 categories in all, did not achieve statistical significance

P. 14, Para. 1, line 289:

Did the participants themselves have high telehealth usability, or did they rather record high telehealth usability on the form they were given?

P. 14, Para. 2, line 300:

Mention the low ratings received for consultation services. Is availability of telehealth  consultants a factor?

P. 14, Para. 3:

line 309: Comment on whether the finding that telehealth usability increases with residency in the eastern or central KSA regions may be linked with the presence of major urban centers with central hospitals that provide substantial telehealth services

line 324 or after: The authors need to point out the significance of this study’s contribution to the existent body of telehealth literature

P. 15, Para. 1, line 328:

Do the authors mean that younger employed participants had higher odds, or employed participants regardless of age had higher odds? If latter interpretation, this statement would seem to be in conflict with prior sentence which asserts that participants of high socioeconomic status had lower odds

P. 15, para. 2:

The authors need to back up these general, 1-word explanations with 2 or so concise examples

P. 15, Para. 3:

line 338: had a higher perception  ->  The reader will wonder, “Had a higher perception than what?” One way to address this matter is to state, “had a higher perception on a scale from X to Y”

line 343: Add a brief sentence explaining Record et al.’s finding

P. 16, Para. 4, line 419:

On P. 14 the authors emphasize the value of asynchronous communication, yet here they emphasize synchronous. Please reconcile.

P. 17, Para. 1, line 17:

The authors cite the urban/rural distinction multiple times in the discussion, yet leave it out of their data collection and analysis. A sentence should be added to the limitations section to the effect: “While we have covered geographic region in our study, our analysis has suggested the value of including urbanicity/rurality in future investigations.”

Grammatical Corrections:

P. 1, Para. 1:

line 19: convenient sample  ->  convenience sample

lines 22, 34: the ordinal regression  ->  ordinal regression; the commonly used  ->  commonly used

[The authors have introduced “the” in many places where it is not called for. Please find a grammatically knowledgeable colleague to help identify and delete these numerous instances.]

P. 2, Para. 3:

line 69: The highest number of the population  ->  the highest proportion of the population

line 73: have given  ->  have generated

P. 2, Para. 4, line 79:

have an invaluable role  ->  have played an invaluable role

P. 3, Para. 5, line 136:

was Adults  ->  was adults

P. 4, Para. 3:

line 162: who proved its reliability where all its attributes had good to excellent reliability  ->  who proved all its attributes had good to excellent reliability 

line 167: subscales scores  ->  subscale score

[The authors should avoid such double plurals throughout the paper]

P. 4, Para. 4:

line 172: The authors need to review subheadings to make sure 1st terms versus subsequent terms demonstrate consistency in capitalization.

line 174: five expertise  ->  five experts

lines 177-8: based on the panel’s feedback.  ->  based on panel feedback.

[Often when removing superfluous “the” terms, the string of terms simplifies.]

P. 4, Para. 5, lines 184-5:

through an online self-administered questionnaire through the Survey Monkey program  ->  through an online self-administered questionnaire using Survey Monkey

P. 5, Paras. 2 and 3:

The Multinomial  ->  Multinomial; The researchers  ->  Researchers;  The respondents  ->  Respondents

P. 8, Para. 1, line 245:

ease of use  ->  having ease of use

P. 8, Para. 2, line 253:

weighted mean differences  ->  weighted mean differences between levels

P. 14, Para. 3:

line 308: Our findings proved that telehealth usability significantly increased with age  ->  Our findings prove that telehealth usability significantly increases with age 

[The findings in the Conclusions need to be phrased in the present tense since the study is recent and the analysis itself is very recent]

line 310: The study also found  ->  The study also finds

P. 15, Para. 1:

lines 326-7: showed that those with high socioeconomic status and older participants  ->  showed that participants with high socioeconomic status and those who are older

P. 15, Para. 2:

line 334: The present study showed  ->  The present study shows

[The use of the term “present study” reinforces the point that assertions in the Conclusions need to be couched in the present tense]

P. 15, Para. 3:

line 339: being women, and chronically ill patients.  ->  being female, and a chronically ill condition.

line 340: which can improve their perceived sense of being helpful, as well as lower the cost  ->  which can heighten their perceived amenability to telehealth, including the lower cost

line 344: among the studied participants  ->  among the participants

[The word “study” appears earlier in the sentence]

P. 15, Para. 4, lines 354-5:

easy to learn and use to facilitate speedy work completion  ->  easy to learn and to use for facilitating speedy work completion

P. 15, Para. 5, lines 363, 369:

The term “trend” is so general and begs for more explanation, i.e., what trend, specifically? Alternative would be to state: “found a lower proportion” or “found a lower mean score”.

P. 16, Para. 2:

line 392: It significantly differed  ->  This relationship significantly differed

line 395: areas rather than in-person  ->  areas, in contrast to in-person

line 398: Similarly, Dawood and Alkadi (2022) showed a similar low-reliability  ->  Dawood and Alkadi (2022) showed a similar low-reliability 

P. 16, Para. 3:

line 405: being women  ->  being female

line 406: could be explained by the comorbidities  ->  could be partially explained by comorbidities 

lines 408-9: Please correct this run-on with use of “the”

line 416: not the older age group.  ->  not only the older age group.

P. 16, Para. 5, line 425:

where  ->  whereas

P. 17, Para. 2:

line 435: The meaning of a population having a high telehealth usability is unclear. Suggest the following modification:   KSA population have high telehealth usability  ->  KSA population record high telehealth usability 

line 442: had a higher perception  ->  had a higher perception on a scale from X to Y

line 443: awkward wording: “A lower trend of high perception” – needs fleshing out

line 445: reliability among  ->  reliability recorded among

P. 17, Para. 3:

lines 446-7: recommend that the need for telehealth future developments to overcome social  ->  recommend that future telehealth developments address

lines 451-2: manuals must be initiated. Moreover, action research projects must be initiated to expand telehealth services based on consumer feedback and shareholder engagement in KSA.  ->  manuals are recommended. Action research projects to expand telehealth services based on consumer feedback and shareholder engagement in KSA are of value [or “are warranted”].

[Authors should avoid issuing imperatives to readers]

P. 17, Para. 5, line 469:

and no Individuals  ->  and no individuals 

P. 18, Ref. 28, line 536:

Spell-out “ISO”

Comments on the Quality of English Language

See enclosed comments.

Author Response

Dear Reviewer 2,

We thank you for your valuable comments and suggestions for improvements.

All the points you raised are addressed (as in the attached file).

English editing was done (the report was attached, done by Grammarly professional premium account of the corresponding author) along with the modified version of the manuscript. 

Regards

Reviewer 3 Report

Comments and Suggestions for Authors

This study was conducted to assess the perceived usability of telehealth for personalized healthcare among Saudi adult population. The authors found that nearly 2/3 of the participants had high telehealth usability. The study could have provided useful information if it did not have two serious methodological flaws.

MAJOR ISSUES

1) The first problem is the use of a convenience sampling, which makes the results not generalizable to the study population. As you can see from Table 1, only 1.5% of the study sample was aged 60 or older, which is clearly an under-representation of the age distribution of the Saudi population. This is due to the use of an online survey and social media. Only those who have had a good command of using the Internet or social media and those who have had good experiences with telemedicine responded. Therefore, the study sample is not representative of the study population. Using samples from different parts of Saudi Arabia does not eliminate this bias, as this overlooking of parts of the population occurs in every region studied. 

2) Another major problem is the use of the TUQ questionnaire. It is mentioned that it is a standard, reliable and valid questionnaire. That's fine, but it's only valid in the original language in which it was developed, not in Arabic, presumably the language in which the questionnaire was used in their study. Although it is mentioned in section 2.2 that the questionnaire has been validated in Arabic, no information on the results of its validation (e.g., Cronbach's alpha for the Arabic version, etc.) is provided in the study. Most of the results come from this questionnaire; without a reliable and valid questionnaire (tool), it is not possible to accept the values provided.

These two issues would jeopardize the external validity of the results obtained from this study. 

MINOR ISSUES

There are also a few minor comments:

3) It is not clear how the sample size was calculated. Section 2.1.1, lines 148-152 provide the parameters necessary for the calculation, but based on the parameters given and equations provided in Reference 31 of the manuscript, the minimum sample size is 379, not 947 or 975. Please elaborate.

4) When reporting the AOR (presumably, the adjusted odds ratio), please provide only the value (to one or two decimal places) followed by the 95% CI. No p-value is required. Examples include line 30 and Table 5.

5) The names of the applications in lines 73-74 are better to be translated into English (for the sake of international readers).

6) Although the "post-COVID-19 era" is presumably one of the concerns of the study, little information is provided on this issue.

Author Response

Dear Reviewer 3,

We thank you for your valuable comments and suggestions for improvements.

All the points you raised are addressed (as in the attached file).

English editing was done (the report was attached, done by Grammarly professional premium account of the corresponding author) along with the modified version of the manuscript. 

Regards

Round 2

Reviewer 2 Report

Comments and Suggestions for Authors

General Comments to Authors:

The paper has undergone substantial improvement. The authors’ explanations have been quite helpful. A few conceptual and grammatical changes remain, then the paper will be ready.

Specific Comments to Authors:

P. 1, Para. 1, line 34:

Thank you for adding the nonsignificant predictors. List is long; would cut it down by removing mention of age, marital status, and nationality. Simplified sentence then starts: “Working status, income …”    [delete “Conversely”]

P. 14, Para. 2, line 328:

Explain that the quotations pertain to asynchronous services. If they pertain to synchronous, delete the quotations since the attempt is being made to support asynchronous services. Make sure that earlier in the paper the distinction between synchronous and asynchronous services is articulated.

P. 15, Para. 2, line 358:

The first sentence is confusing because individuals of high socioeconomic status are generally those who are employed. Does the article referenced make an additional clarifying distinction so that high SES individuals and those who are employed can differ on telemedicine use without generating a seeming contradiction?

Grammatical Corrections:

P. 1, Para. 1:

lines 19-20: of 975 general adult population in KSA.  ->  of 975 adults from the general  population in KSA. 

lines 22-3: and were investigated using the multinomial  ->  and investigated using  multinomial

line 24: most frequently used  ->  most frequently used services

P. 3, Para. 4:

line 123: reaching 12.01  ->  has reached 12.01

line 129: which needs help to solve the problem.  ->  which imposes barriers towards adoption.

P. 3, Para. 5, line 137:

offers the context  ->  offers context 

P. 4, Para. 2 Subheading, line 154:

The subheading formatting remains inconsistent. Please capitalize each of the main words in the 2-digit subheadings here and for the rest of the paper, removing the colon or period at the end (3-digit subheadings look O.K.). Here:

Study design: Population-based cross-sectional design.  ->  Study Design: Population-based Cross-sectional Design

P. 5, Para. 4, line 231:

(p>0.05)  ->  (p>=0.05) 

P. 5, Para. 5, line 239:

The respondents  ->  Respondents

P. 9, Para. 2, line 276:

mean differences of the total  ->  mean differences between levels of the total

P. 12, Table 4 Subheading:

Left orient the newly added terms “Commonly   used”

P. 14, Para. 1:

lines 312-3: participants had high telehealth usability,  ->  participants rated telehealth as having high usability

line 323: This could be attributed to  ->  This finding matches with

P. 14, Para. 3, line 337:

This may  ->  This finding may

P. 16, Para. 3:

line 432: It significantly  ->  This relationship significantly

line 433: has the lowest scores.  ->  has the lowest scores (Table 4).

P. 16, Para. 4:

line 446: by the comorbidities  ->  by comorbidities

line 448: The residential area may also affect the internet  ->  Residential area may also affect  internet 

line 449: connection and interaction with the healthcare providers, affecting the participants’  ->  connections and interactions with healthcare providers, affecting participants’ 

P. 17, Para. 1:

line 459: It explored  ->  It found

line 468: [21, 19,45].  ->  [19, 21, 45]. 

P. 17, Para. 2:

line 471: where most  ->  whereas most

line 481: which proved variations  ->  which provided variations

line 482: this highlights  ->  this heterogeneity highlights

P. 17, Para. 3:

line 487: and more than half have initiated  ->  with more than half initiating

line 496: reliability preconception among   ->    reliability observed among 

Comments on the Quality of English Language

See attached comments.

Author Response

Dear Reviewer 2,

Thanks for your critical revision; your comments help us a lot in improving our work (a reply document is attached).

Perceived Telehealth Usability for Personalized Healthcare Among Adult Population in Saudi Arabia: A Cross-Sectional Study in the Post-COVID-19 Era

Reply to reviewer 2

Comment

Reply

Section-Page

Specific Comments to Authors:

P. 1, Para. 1, line 34:Thank you for adding the nonsignificant predictors.List is long; would cut it down by removing mention of age, marital status, and nationality.Simplified sentence then starts: “Working status, income …” [delete “Conversely”]

Done as per your comment, highlighted in purple.

P. 1, Para. 1, line 34

P. 14, Para. 2, line 328:Explain that the quotations pertain to asynchronous services. If they pertain to synchronous, delete the quotations since the attempt is being made to support asynchronous services.Make sure that earlier in the paper, the distinction between synchronous and asynchronous services is articulated.

The quotations are moved to the end of the paragraph to support the issue raised by the evidence, which is highlighted in red. A brief description was added in the introduction, distinguishing between synchronous and asynchronous telehealth services, highlighted in purple.

P. 14, Para. 2, Lines 329-334.Page 2, Para 2, Lines 53-54

P. 15, Para. 2, line 358:The first sentence is confusing because individuals of high socioeconomic status are generally those who are employed. Does the article referenced make an additional clarifying distinction so that high SES individuals and those who are employed can differ on telemedicine use without generating a seeming contradiction?

Thanks for your critical revision; it has been corrected and highlighted in purple. I returned to the original study and discovered that it is a type. Thanks

P. 15, Para. 2, line 358

Grammatical Corrections:

P. 1, Para. 1:lines 19-20: of 975 general adult population in KSA. -> of 975 adults from the general population in KSA.

Done and highlighted in purple

lines 22-3: and were investigated using the multinomial -> and investigated using multinomial

Done, deleted “were”

line 24: most frequently used -> most frequently used services

Done and highlighted in purple

P. 3, Para. 4:line 123: reaching 12.01 -> has reached 12.01

Done and highlighted in purple

line 129: which needs help to solve the problem. -> which imposes barriers towards adoption.

Done and highlighted in purple

P. 3, Para. 5, line 137:offers the context -> offers context

Done, deleted “the”

P. 4, Para. 2 Subheading, line 154:The subheading formatting remains inconsistent. Please capitalize each of the main words in the 2-digit subheadings here and for the rest of the paper, removing the colon or period at the end (3-digit subheadings look O.K.). Here:Study design: Population-based crosssectional design. -> Study Design: Population-based Cross-sectional Design

Done and highlighted in purple

P. 5, Para. 4, line 231:(p>0.05) -> (p>=0.05)

Done and highlighted in purple

P. 5, Para. 5, line 239:The respondents -> Respondents

Done and highlighted in purple

P. 9, Para. 2, line 276:mean differences of the total -> mean differences between levels of the total

Done and highlighted in purple

P. 12, Table 4 Subheading:Left orient the newly added terms “Commonly used.”

Done

P. 14, Para. 1:lines 312-3: participants had high telehealth usability, -> participants rated telehealth as having high usability

Done and highlighted in purple

line 323: This could be attributed to - > This finding matches with

Done and highlighted in purple.As per your new comment in round 2, it shifted to line 331.

P. 14, Para. 1:Lines 331

P. 14, Para. 3, line 337:This may -> This finding may

Done and highlighted in purple.

P. 16, Para. 3:line 432: It significantly -> This relationship significantly

Done and highlighted in purple.

line 433: has the lowest scores. -> has the lowest scores (Table 4).

Done and highlighted in purple.

P. 16, Para. 4:line 446: by the comorbidities -> by comorbidities

Done, deleted “the”

Line 447

line 448: The residential area may also affect the internet -> Residential area may also affect internet

Done, deleted “the”

Line 449

line 449: connection and interaction with the healthcare providers, affecting the participants’ -> connections and interactions with healthcare providers, affecting participants’

Done, deleted “the”

Line 450

P. 17, Para. 1:line 459: It explored -> It found

Done and highlighted in purple.

line 468: [21, 19,45]. -> [19, 21, 45].

Space is fixed

P. 17, Para. 2:line 471: where most -> whereas mostline 481: which proved variations - > which provided variationsline 482: this highlights -> this heterogeneity highlights

Done and highlighted in purple.

P. 17, Para. 3:line 487: and more than half have initiated - > with more than half initiatingline 496: reliability preconception among - > reliability observed among

Done and highlighted in purple.

Regards

Reviewer 3 Report

Comments and Suggestions for Authors

Thank you very much for revising the manuscript.

Author Response

Dear Reviewer 3,

Comment: I believe the revised version can be published in your prestigious Journal.

Reply: We are grateful for your critical revision and valuable comments during round one, which helped us a lot in improving our work.

Thanks and regards